

# Transition to El Niño conditions in the eastern tropical Pacific in October 2015

Lothar Stramma[1], Tim Fischer[1], Damian S. Grundle[1], Gerd Krahmann[1], Hermann W. Bange[1] and Christa A. Marandino[1]

[1]GEOMAR Helmholtz Centre for Ocean Research Kiel, Düsternbrooker Weg 20, 24105 Kiel, Germany
*Correspondence to*: L. Stramma (lstramma@geomar.de)

**Abstract.** A strong El Niño developed in early 2015. Measurements from a research cruise on the RV *Sonne* in October 2015 near the equator east of the Galapagos Islands and off the shelf of Peru, are used
to investigate changes related to El Niño in the upper ocean in comparison with earlier cruises in this region. At the equator at 85°30'W, a clear temperature increase leading to lower densities in the upper 350 m, despite a concurrent salinity increase from 40 to 350 m, developed in October 2015. Lower nutrient concentrations were also present in the upper 200 m, and higher oxygen concentrations were observed between 40 and 130 m. Except for the upper 60 m at 2°30'S, however, there was no obvious
increase in oxygen concentrations at sampling stations just north (1°N) and south (2°30'S) of the equator at 85°30'W. In the equatorial current field, the Equatorial Undercurrent (EUC) east of the Galapagos Islands almost disappeared in October 2015, with a transport of only 0.02 Sv in the equatorial channel between 1°S and 1°N, and a weak current band of 0.78 Sv located between 1°S and 2°30'S. Such near-disappearances of the EUC in the eastern Pacific seem to occur only during strong El
Niño events. Off the Peruvian shelf at ~9°S, where the sea surface temperature (SST) was elevated, upwelling was modified, and warm, saline and oxygen rich water was upwelled. Despite some weak El Niño related SST increase at ~12 to 16°S, the upwelling of cold, low salinity and oxygen-poor water was still active at the easternmost stations at three sections at ~12°S, ~14°S and ~16°S, while further west on these sections a transition to El Niño conditions appeared. Although in early 2015 the El Niño
was strong and in October 2015 showed a clear El Niño influence on the EUC, in the eastern tropical Pacific the measurements only showed developing El Niño water mass distributions. In particular the



oxygen distribution indicated the ongoing transition from 'typical' to El Niño conditions progressing southward along the Peruvian shelf.

# 1 Introduction

Past El Niño events have been observed to have different local occurrences and parameter distributions in recent years. In contrast to the cold tongue, or Eastern Pacific (EP), El Niño events which develop in the eastern Pacific, there has been evidence of an increased occurrence of El Niño events in the central Pacific called Central Pacific (CP) El Niño or 'El Niño Modoki' (Dewitte et al., 2012). For a typical CP El Niño, the largest SST increase occurs at the equator between 130°W and 160°E while cooling appears off the shelf of Peru, while for the EP El Niño the SST increases at the equator east of 180°W to South America and southward along the South American coast to Chile (e.g. Dewitte et al., 2012). Different indices exist to describe the El Niño status. The NINO 1+2 index is the temperature difference compared to the 1982-2005 climatological cycle in the eastern tropical Pacific (0-10°S, 80-90°W), and is close to the region of the measurements used here. The Oceanic Niño Index (ONI) has become a standard for identifying El Niño and La Niña events. It is a running 3-month mean sea surface temperature (SST) anomaly for the Niño 3.4 region (i.e., 5°N-5°S, 120-170°W) related to the 1971-2000 base period. Events are defined as 5 consecutive overlapping 3-month periods at or above the 0.5°C anomaly for warm El Niño events, and at or below -0.5°C anomaly for cold La Niña events. The strongest El Niño events since 1950 were observed in the years 1982/83 and 1997/98, the latter also referred to as 'the climate event of the twentieth century' (Changnon, 2000). There is climate modelling evidence for a doubling in the occurrences of extreme El Niño events in the future in response to greenhouse warming (Cai et al., 2015). In early 2015, an El Niño with strength similar to the 1997/98 El Niño developed. Sea surface temperature anomalies were strongest along the equator and the tropical North Pacific, while the development of a temperature anomaly in the eastern tropical Pacific off Peru was, according to NOAA's 'ENSO diagnostic discussion archive', strong in April and May, then weakened and intensified again August to October 2015.

El Niño dynamics modulate near-surface temperature, salinity and density, as well as the mixed layer depth, oxycline depth, and the vertical extent of the low oxygen layer (e.g. Fuenzalida et al., 2009). In





the Pacific Ocean, El Niño events are a prominent feature of climate variability with global climate impacts. In the eastern Pacific, El Niño-Southern Oscillation (ENSO) variability is most pronounced along the equator and the coasts of Ecuador and Peru (Wang and Fiedler, 2006). Weaker trade winds during El Niño conditions result in a weaker equatorial circulation with a generally observed weakening

or disappearance of the Equatorial Undercurrent (EUC) (Johnson et al., 2002). During the height of an El Niño event, the EUC episodically disappears in the western and central Pacific (Firing et al., 1983; McPhaden et al., 1990; Johnson et al., 2000; Izumo, 2005), while in the eastern Pacific, episodic disappearance of the EUC seems more rare and was only observed during strong El Niño events (Halpern, 1987; McPhaden and Hayes, 1990; Seidel and Giese, 1999; Johnson et al., 2000; Izumo,

2005). El Niño events lead to a pronounced eastward extension of the west Pacific warm pool, and to a development of atmospheric convection, and hence a rainfall increase, in the usually cold and dry eastern Pacific (Cai et al., 2015).

In the Eastern Tropical South Pacific (ETSP), a subsurface low-oxygen zone exists with a pronounced minimum in oxygen at ~100 to 500 m depth and is referred to as an oxygen minimum zone (OMZ) or

oxygen deficient zone (ODZ). This ODZ is suboxic (oxygen concentrations below ~4.5-10.0 µmol kg$^{-1}$; e.g. Karstensen et al., 2008; Stramma et al., 2008). In suboxic regions nitrate and nitrite become involved in respiration processes such as denitrification or anammox (e.g. Kalvelage et al., 2013). In the eastern equatorial Pacific the oxygen content has been shown to increase during El Niño events in the upper 300 to 350 m in the equatorial channel (e. g. Fuenzalida et al., 2009; Czeschel et al., 2012), as

well as off the Peruvian coast as a result of circulation changes (Helly and Levin, 2004). Coastal winds during El Niño events are usually upwelling-favourable. Coastal warming during El Niño is caused by downwelling Kelvin waves generated by mid-Pacific westerly wind anomalies that deepen the eastern thermocline and allow warming to occur, independent of the local winds (Kessler, 2006). Consequently, during El Niño events the upwelled water off Peru is warmer, more oxygen replete and less nutrient

rich. El Niño, in general, results in a depressed thermocline and thus reduced rates of macronutrient supply and primary production (Pennington et al., 2006) off Peru. In case of strong El Niño events when the oxygen concentration above the shelf bottom increases from about zero to >40 µmol kg$^{-1}$ the sediments respond with tremendous changes of ecological state (Gutiérrez et al., 2008). At a time-series



station at ~12°S, 77°30'W off Lima from 1996 to 2010 for temperature, salinity, density, oxygen and nutrients, the influence of El Niño - especially the strong 1997/1998 El Niño - is clearly visible with higher temperature, salinity and oxygen, and lower density, nitrite, silicate and phosphate (Graco et al., 2016).

Here we use measurements from an *RV Sonne* research cruise in October 2015 (Fig. 1) east of the Galapagos Islands from an ADCP section from the Ecuadorian shelf to 1°N, 85°30'W, from a section across the equator and from 4 sections off the Peruvian shelf to investigate changes in the upper ocean related to the strong 2015 El Niño in comparison with earlier cruises in this region. The aim is to unravel the progress of the transition to El Niño conditions in the eastern Pacific several months after
the start of the El Niño.

## 2 Data sets

In October 2015 an *RV Sonne* transit cruise (So243; 5 to 22 October 2015) from Guayaquil, Ecuador to Antofagasta, Chile was carried out (Fig. 1) (short cruise report available at: https://www.ldf.uni-
hamburg.de/sonne/wochenberichte/wochenberichte-sonne/so242-243/so243-scr.pdf), which allowed us to investigate possible El Niño signals at the equator near 85°30'W and off the shelf of Peru at sections perpendicular to the shelf at ~9°S, ~12°S, ~14°S and ~16°S.

A Seabird CTD system with a GO rosette with 24 x 10 L water bottles was used for water profiling and discrete water sampling. The CTD system was used with double sensors for temperature, conductivity
(salinity) and oxygen. The dual CTD temperature sensors calibrated by the manufacturer are compared during the cruise that the deviation is less than 0.002°C and the accuracy of the temperature measurements is estimated to be 0.002°C or better. The CTD salinity calibration with salinometer salinity samples resulted in a r.m.s. uncertainty of 0.0011. The CTD oxygen sensors were calibrated with oxygen measurements obtained from discrete samples from the rosette applying the classical
Winkler titration method, using a non-electronic titration stand (Winkler, 1888; Hansen, 1999). The r.m.s. uncertainty of the CTD oxygen sensor calibration of cruise So243 was determined as ±0.8 μmol kg$^{-1}$. Oxygen concentrations of less than 3 μmol kg$^{-1}$ are not resolved by Winkler titration and values

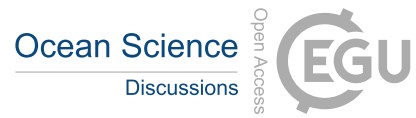

below 3 μmol kg$^{-1}$ were used as 0 μmol kg$^{-1}$ for the sensor calibration as the H$_2$S smell of the water of related rosette bottles indicated 0 μmol kg$^{-1}$.

Nutrients were measured on-board with a QuAAtro auto-analyzer (Seal Analytical). Nitrite (NO$_2^-$), nitrate (NO$_3^-$), phosphate (PO$_4^{3-}$) and silicid acid (Si(OH)$_4$ referred to silicate hereinafter) were

measured with an analytical precision of 5.5%, 1.3%, 0.4% and 0.5% respectively. The N:P ratio used here was computed as N:P= (NO$_3^-$ + NO$_2^-$): PO$_4^{3-}$ .

Two vessel mounted Acoustic Doppler Current Profilers (ADCP) were used to record ocean velocities in October 2015: an RDI OceanSurveyor 75 kHz ADCP with 8 m bin spacing provided the velocity distribution to ~650 m depth, while a 38 kHz ADCP with 32 m bin spacing provided velocity profiles

down to ~1300 m depth. During the entire cruise the navigation data was of high quality. Due to the interest in the upper ocean the higher resolution 75 kHz ADCP is used here.

Earlier crossings of the equator (Table 1 and Fig. 1) were accomplished in March/April 1993 on RV *Knorr* (Tsuchiya and Talley, 1998), in February 2009 on RV *Meteor* (Czeschel et al., 2011) and in November 2012 on RV *Meteor* (Stramma et al., 2013) at 85°50'W. Sections across the Peruvian shelf

between 9°S and 16°S were made during RV *Meteor* cruise M91 in December 2012 (Czeschel et al., 2015; Bange, 2013). Measurement accuracies during these cruises were similar as in October 2015 and the details are described in the related literature. In contrast to October 2015, the CTD stations in 1993, 2009 and 2012 were not carried out at 2°30'S, 85°30'W but at 2°20'S and 2°40'S at 85°50'W and these two stations were combined for a mean profile at 2°30'S. The sections across the equator and off the

Peruvian shelf were not at identical geographical coordinates, but we expect that the offset is small compared to the differences measured.

## 3 The El Niño in 2015

The SST anomaly for 27 September 2015 to 24 October 2015 was strong along the equator to the South

American continent and southward off the Peruvian coast (Supplement Fig. S1). The NINO 1+2 index was high at +2.52 in October 2015 (Table 1), hence the 2015 El Niño is a clear EP El Niño. Different to the typical EP El Niño SST distribution in fall 2015, a strong and prominent SST increase was also



present along Central America and in the eastern North Pacific at 20-25°N. This feature, also known as 'The Blob', is an unrelated positive temperature anomaly which developed in 2013 in the Gulf of Alaska and progressed along the North American continent to the 20-25°N region in mid-2015 (Kintisch, 2015).

5 The only El Niño events since 1950 with an October maximum ONI of more than 1.7, or an overall maximum of 2.0 or larger, are the 1972/73, 1982/83 and the 1997/98 El Niños. In early 2015 the ONI index was even larger than the ONI index of these 3 large El Niño events, while in October 2015 it was at a similar strength as the three earlier strong El Niños (Fig. 2). Accordingly, the 2015 El Niño has to be listed as one of the 4 strongest El Niño's since 1950.

## 4 The equatorial region east of the Galapagos Islands

## 4.1 The hydrographic variability
### 4.1.1 Background information

The hydrographic distribution in the eastern equatorial Pacific is influenced by a seasonal cycle as well as El Niño related cycles. At 110°W in the eastern Pacific west of the Galapagos Islands, relationships between zonal velocity, temperature, and salinity in the EUC are all evident in the seasonal cycle. The EUC peaks in strength around April/May, when it also surfaces (Johnson et al., 2002). The thermocline is extremely sharp and shallow. The equatorial spreading of the thermocline associated with the EUC is

noticeably stronger during April than in October, when in April equatorial SST is lowest and the South Equatorial Current (SEC) is strongest. The laterally isolated salinity maximum within the thermocline just south of the equator is strongest when the EUC velocity is at its greatest (Johnson et al., 2002), and this is also visible in the sea surface salinity (Supplement Fig. S2) from the MIMOC climatology (Schmidtko et al., 2013). Between austral fall and winter the ODZ core in the eastern South Pacific at

the equator intensifies from 16 to 15 µmol $O_2 L^{-1}$ for the mean oxygen concentration and from 8 to 5 µmol $O_2 L^{-1}$ for the minimal oxygen concentration (Paulmier and Ruiz-Pino, 2009).



Weaker trade winds during El Niño conditions result in a weaker equatorial circulation while stronger trade winds during La Niña conditions lead to a stronger equatorial circulation (Johnson et al., 2002). During La Niña, the current system at 110°W is spun upwards when compared to El Niño. The cold tongue located in the eastern tropical Pacific is quite weak during El Niño. Surface salinities are

generally fresher during El Niño than during La Niña, a feature which is at least partially a product of increased local precipitation associated with the eastward migration of warm sea surface temperatures and convection, and partly a result of the reduced trade winds (Johnson et al., 2002). For the 1996-1998 El Niño-La Niña cycle, a very fresh mixed layer in the eastern equatorial Pacific late in the El Niño, and a reduction of the strength of the meridional equatorial salinity gradient within the pycnocline to one

third of the usual value during the El Niño was observed (Johnson et al., 2000). Hence, during El Niño events higher salinity should be expected near the pycnocline.

### 4.1.2 Observations for the 2015 El Niño

SST anomalies for the period 27 September to 24 October 2015 showed an SST anomaly of 2.0-2.5°C

at 85°30'W at and south of the equator, and of 1.5-2.0°C just north of the equator (Supplement Fig. S1). In the upper 100 m of the water column, oxygen, temperature, salinity and density profiles at the equator on the ~85°30'W meridian (Fig. 3) reveal differences between March 1993, February 2009, November 2012 and the El Niño of October 2015. It is important to note that 1993 was not defined as an official El Niño year as only 4, instead of 5, consecutive overlapping 3-month periods were at or

above the 0.5°C anomaly. However, in March and April 1993 the NINO 1+2 index reached +0.65 and +0.97 (Table 1) and had El Niño like SST anomalies. To this end, we will refer to March 1993 as 'El Niño-like' hereinafter.

In February 2009 the ONI index was for the fourth and last month -0.5 or less, therefore conditions were similar to a weak La Niña event and we will refer to it as 'La Niña-like' hereinafter. In February

2009, in the upper 100 m at the equator at ~85°30'W the oxygen and temperature were lowest and the density highest compared to the other 3 periods (Fig. 3), representing an expected La Niña parameter distribution. The hydrographic profiles in the neutral ONI period in November 2012 mainly lay between the El Niño profiles for March 1993 and October 2015, and the La Niña-like profiles in February 2009.

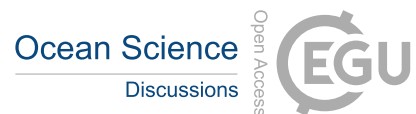

The November 2012 were somewhat closer to the February 2009 profiles. The El Niño profiles in October 2015 and the El Niño-like profiles in March 1993 showed slightly higher oxygen concentrations and temperature, and lower density in the upper 100 m in comparison to November 2012 and February 2009 (Fig. 3). In October 2015 the salinity compared to the 3 other years was lowest in a

deep thermocline in the upper 40 m as expected for the surface layer during an El Niño event because of the reduced equatorial upwelling. In contrast, a weak salinity maximum was located below 40 m as expected near the pycnocline as saline warm water progresses from the western Pacific eastward during El Niño. In the profiles to 400 m depth, the higher temperature and salinity, and lower density reached down to ~350 m in October 2015, while the oxygen profile below 130 m merges with the profiles from

the other measurement periods (Fig. 4).

The strong thermocline/pycnocline of the eastern tropical Pacific is also a strong nutricline. A consistent general pattern is that nitrate and phosphate increase with depth to ~500 m with a slight maximum at intermediate depths, while silicate continues to increase with depth (Fiedler and Talley, 2006). The vertical distribution of nutrients at the equator at ~85°30'W shows lower nitrate, phosphate and silicate

concentrations in the upper 200 m in October 2015 as well as in the El Niño-like year 1993 in comparison to the 2009 and 2012 concentrations (Fig. 5). A primary nitrite maximum (PNM) usually occurs in the lower euphotic zone, and results from nitrite excretion by phytoplankton and/or a decoupling of ammonia and nitrite oxidation (i.e. higher rates of ammonia vs. nitrite oxidation; Fiedler and Talley, 2006; Lomas and Lipschultz 2006). Nitrite is close to undetectable below 100 m depth (Fig.

5). In 1993 and 2015, however, the PNM was located ~25 m deeper and maximum nitrite concentrations were considerably higher. This reflects the deeper pycnocline in El Niño years. Furthermore, the deep thermocline and pycnocline in October 2015 indicate that equatorial upwelling was reduced. According to the upper ocean hydrographic and nutrient distribution in October 2015, a clear El Niño situation had adjusted at the equator at ~85°30'W.

At 1°N, ~85°30W in the 50-300 m layer, salinity and temperature were higher and the density lower in October 2015 than at the 3 other times, however, the oxygen was not significantly higher during this time (Supplement Fig. S3). At 2°30'S, ~85°30'W in the 50 to 250 m layer, the salinity, temperature and oxygen were lower and the density higher in the La Niña-like month of February 2009 than at the other



three years. The temperature at 2°30'S in the El Niño of October 2015 was higher in the 50 to 250 m depth range than in the other 3 years and salinity showed slightly higher values at ~50 to 100 m and 150 to 250 m depth. The oxygen concentration was slightly higher for both El Niño profiles in March 1993 and in October 2015 in the upper ~60 m compared to 2009 and 2012 (Supplement Fig. S4). Earlier selected measurements at 4°S, 85°W showed a clear oxygen increase from the near-surface to ~350 m depth for El Niño years (Czeschel et al., 2012). Hence, the El Niño influence on the water mass distribution was still weak at 2°30'S in October 2015 and only developing in the upper ocean.

## 4.2 Circulation observations

### 4.2.1 Background information

The EUC, which carries oxygen rich water towards the ODZs of the eastern Pacific, has a seasonal cycle, with a transport peak in the eastern Pacific at 95°W in April/May of ~30 Sv and a minimum in October/November of little less than 15 Sv (Johnson et al., 2002; their Figure 17). The Galapagos Islands form a barrier to the EUC which causes it to bifurcate into a shallow/southern core centered at ~50 m depth (EUCs) and a deeper/northern core centered at ~150 m depth (EUCd) (Karnauskas et al., 2010). Results from a climate model for 110°W show an increase in the surface eastward EUC current during austral fall, while during other seasons the EUC is at deeper depth (Cravatte et al., 2007). ROMS (Regional Ocean Model System) model runs east of the Galapagos Islands display weaker transports which are variable, depending on the boundary conditions provided by different ocean general circulation models (OGCMs). The modelled EUC transports at 86°W between 2°N and 2°S at 200 m depth for February/March and October/November are ~10-12 Sv and ~7-8 Sv for OCCAM, ~6-8 Sv and 4-5Sv for SODA, and ~5.5 Sv and 7-8 SV for ORCA (Echevin et al., 2011).

During El Niño events the equatorial circulation system of the Pacific weakens, and the eastward flowing EUC has been found to decay (Kessler and McPhaden, 1995, Johnson et al., 2002) and partially reverse, e.g. during the 1982-1983 El Niño at 159°W in September 1982 (Firing et al., 1983). During the 1997/98 El Niño shipboard current measurements showed that the EUC virtually disappeared across much of the Pacific basin, associated with the weakening or even the reversal of the equatorial pressure gradient within the pycnocline (Johnson et al., 2000). For the 1982-1983 El Niño there seems to be a



strong time delay for the EUC weakening. In September 1982 at 159°W, the EUC reversed (Firing et al., 1983), however, in November 1982 at 95°W, the EUC was strong in November 1982 before being replaced by a westward jet in May 1983 (Hayes et al., 1986).

**4.2.2 Observations for the 2015 El Niño**

We are not aware of any El Niño related EUC variability observations east of the Galapagos Islands. Model results (Montes et al., 2011) for El Niño periods show a weakening and southward shift of the EUC branches east of the Galapagos Islands where the EUC flows at shallower depths and is associated with lighter water. The direct velocity observations in October 2015 on the diagonal section from the

Ecuadorian shelf to 1°N, 85°30'W, show only a weak signature of the zonal EUC in the upper 100 m located mainly at, and south, of the equator (Fig. 6a). The weak transport of the EUC is 0.01 Sv between 1°S and 1°N, and 0.29 Sv between 2°30'S and 1°S. The westward flow in the upper 200 m is mainly connected to a northward flow direction (Fig. 6b) north of 1°S. This north-westerly flow indicates the flow of oxygen-poor water from the ODZ off the South American continent to the west

near the equator.

The direct velocity observations on the meridional section at 85°30'W between 1°N and 2°30'S and the diagonal continuation to 5°S, 84°16'W, again show the weak signature of the EUC in the upper 100 m located mainly at, and south, of the equator north of 2°30'S in October 2015 (Fig. 7d). The EUC transport in the upper 300 m is 0.02 Sv between 1°S and 1°N and 0.78 Sv between 2°30'S and 1°S

(Table 2). The stronger eastward flow component in the upper 200 m between 3°S and 4°S might be a combination of the EUC and the Southern Subsurface Countercurrent (SSCC; also called Tsuchiya Jet), as described for El Niño periods east of the Galapagos Islands in model results (Montes et al., 2011). The eastward flow near 2°S below 200 m is the South Intermediate Countercurrent (SICC).

The strongest EUC in our four measurement periods occurred at the end of March 1993, with 12.77 Sv

between 1°S and 1°N in the upper 300 m (Table 2; Figure 7a). This measurement at the end of March was close to the time of eastern Pacific EUC peak transport in April/May. In addition, it was at the beginning of an El Niño-like phase, where warmer, oxygen-rich water is transported from the western to the eastern tropical Pacific and could enhance the eastward flow component.



In February 2009 the EUC transport between 1°S and 1°N was weak (3.55 Sv), although it occurred only two months before the time of the seasonal EUC peak transport. As previously described, February 2009 was at the end of a short La Niña-like period with an ONI index of -0.7, and the low EUC transport might be related to a generally weak eastward transport of warm western equatorial Pacific

water during La Niña. On a cruise approximately 1.5 months later in March to April 2009 between the Galapagos Islands and Ecuador, a region of possible strong cross-hemispheric exchange was observed immediately to the east of the Galapagos, where a shallow (200 m) 300 km wide north-eastward surface flow transported 7 to 11 Sv (Collins et al., 2013). This north-eastward flow might have weakened the EUC transport at and south of the equator. The two diagonal sections in March/April 2009 crossed the

85°50'W section at ~1°50'S and 2°30'S and, similar to the February 2009 measurements, showed a 50 m depth eastward and westward flow at 1°50'S and 2°30'S, respectively, and a westward flow at both of these latitudes at 200 m depth. In contrast to the velocity distribution in March 1993, November 2012 and October 2015 (Fig. 7), the eastward flow component in the upper 200 m south of 2°30'S almost disappeared in February 2009.

The EUC transport in November 2012 at 85°50'W between 1°S and 1°N was 10.78 Sv in the upper 300 m (Table 2). The months before these measurements had no large ONI index values and should represent the non-El Niño EUC transport in this region for November. The transport of 10.78 Sv in November at 85°50'W is less than the November minimum at 95°W of ~15 Sv, and seems to be a reasonable estimate east of the Galapagos Islands, as the EUC transport decreases in the eastern Pacific.

The core of the EUC below 200 m is quite deep and agrees with the seasonal cycle where the EUC should be located at deeper depth in austral spring.

## 5 The upwelling region off Peru

### 5.1 Background information

Off Peru a highly productive year-round upwelling system is located between 4 and 16°S (Chavez and

25 Messié, 2009). The SST off of Peru measured at six locations between 5°S and 12°S over a period of six years shows a seasonal cycle of 2 to 3°C amplitude with the largest SST near March and the minimum near October (Montes et al., 2011; their Fig. 4). This seasonal cycle is also visible in the



MIMOC climatology for 9°S and 12°30'S (Supplement Fig. S2). The time series station at ~12°S, 77°30'W shows a seasonal cycle of about 20 m displacement for the 15°C isotherm, the oxycline depth and the upper boundary of the ODZ (Graco et al., 2016). Seasonal eddy fluxes are described along the coast of Peru with the largest signal at approximately 15°S with a peak during the Austral winter

(Vergara et al., 2016). Since the 1950s, an SST decline corresponding to an increase of upwelling was observed off Peru (Gutiérrez et al., 2011; Sydemann et al., 2014). The typical nutrient distribution along a cross-shelf section at 12°S in December 2012 shows elevated phosphate concentrations in the surface waters near the coast whereas nitrate was depleted in the water column and the near surface waters close to the coast (Kock et al., 2016; their Fig. 3).

Conditions which develop along the coast of Ecuador, Peru and northern Chile during El Niño events include a strengthening of the poleward flow along the coast of Peru, persistent deepening of the thermocline, reducing or even reversing the prevailing upwelling induced land-sea temperature gradient, and a southward shift in the position of the ITCZ which brings heavy precipitation to normally arid regions (Strub et al., 1998). A reduction in coastal cloud cover due to warmer water next to the coast

may enhance insolation and reduce atmospheric pressure over land, maintaining the pressure difference and winds over the coast. As a result, upwelling-favourable winds are not greatly reduced when El Niño conditions are observed in the ocean (Strub et al., 1998). In general, upwelling-favourable winds and upwelling continue during El Niño events, and water continues to be drawn from 50-100 m depth to the surface layer, but the thermocline and nutricline are displaced downward and thickened, so that

upwelling during El Niño brings only warm and nutrient-poor water to the surface (Strub et al., 1998). The intensity of the upwelling appears to be determined by an interplay between alongshore, poleward advection, and wind intensity, but also by the cross-shore geostrophic flow and distribution of the water masses on a scale of 1000 km or more (Colas et al., 2008). In relation to the downward displacement of the thermocline and nutricline, the oxycline is displaced downward. For the 1997/98 El Niño event,

Helly and Levin (2004) described a depression of the upper layer of the ODZ by 100 m reducing the ODZ area off Peru and northern Chile by 61%.



## 5.2 Observations for the 2015 El Niño

The SST anomalies for the period 27 September to 24 October 2015 (Supplement Fig. S1), showed a strong SST anomaly of 1.5-2.0°C between 8°S and 14°S and a weaker anomaly of 0.5-1.5°C between 14°S and 20°S. Differing hydrographic distributions were measured off Peru at ~9°S in December 2012

with a neutral ONI status, and in October 2015 with the strong El Niño. In the entire upper 300 m at ~9°S, temperature, salinity (Supplement Fig. S5 and S6) and oxygen (Fig. 8) were higher in October 2015 than in December 2012. In contrast to the typical seasonal cycle which is characterized by lower SST in October than in December, the SST at 9°S was higher in October 2015 than in December 2012 as a result of the El Niño related SST increase. Higher upper water column temperatures in October

2015 also correlated to lower densities in the upper 300 m (as can be seen from the selected isopycnals in Fig. 8) despite the concurrent influence on density from the salinity increase. Accordingly the density changes are temperature dominated. In December 2012 there was strong upwelling at ~9°S with the <5 $\mu$mol kg$^{-1}$ $O_2$ layer located below ~30 m depth, while in October 2015 this low oxygen layer was only found below 240 m depth. The October 2015 nutrient profiles obtained from shelf stations at ~9°S with

water depths of little more than 100 m (not shown), highlight that nitrate, phosphate and silicate concentrations were lower, and nitrite concentrations were higher in comparison to profiles from the same location in December 2012, as would be expected for El Niño periods. Although the isopycnals and parameter distribution shows that upwelling at 9°S was occurring in October 2015, it is clear that warmer, saline and oxygen replete water was being upwelled, and that the contribution of oxygen

deplete and nutrient rich water was strongly reduced.

At ~12°S the measured oxygen distributions for December 2012 and October 2015 are quite similar in the upwelling region at the easternmost station pair with oxygen concentrations of less than 5 $\mu$mol kg$^{-1}$ (Fig. 9). The oxygen concentration between the isopycnals $\sigma_\theta$= 25.6 and 25.8 kg m$^{-3}$ was even lower in October 2015 than in December 2012 in the upwelling region east of ~77°30'W (Fig. 9). However,

below the oxycline below 50 m depth, temperature, salinity and oxygen concentrations (Supplement Fig. 7f) were higher in October 2015 than in December 2012 and indicate the transition to El Niño conditions. The seasonal signal in the time-series station at ~12°S, 77°30'W shows a shallower 15°C isotherm and oxycline depth of about 20 m in October than in December (Graco et al., 2016), hence the




deeper oxycline in October 2015 compared to December 2012 is not a seasonal signal but an El Niño influence. The nutrient distribution at the shelf at ~12°S (Supplement Fig. S7) also shows El Niño influence with lower phosphate and silicate in October 2015 than in December 2012. This is in agreement with the observed increase in temperature and salinity, and the lower phosphate and silicate

at the time series station at ~12°S during the strong 1997/1998 El Niño (Graco et al., 2016). However, nitrate is lower and nitrite higher in December 2012 than in October 2015, with nitrite reaching 5.4 µmol L$^{-1}$ at 75 m in December 2012 (Supplement Fig. S7) consistent with the observations of Kock et al. (2016). At the depths of the high nitrite concentrations very low oxygen concentrations of less than 2 µmol kg$^{-1}$ were measured. Under low oxygen conditions incomplete nitrification, incomplete

denitrification, or a combination of both, can result in accumulations of nitrite (e.g. Codispoti and Christensen 1985; Gruber 2008; Brockmann and Morgenroth 2010), as was likely the case during 2012. The higher oxygen concentrations in the ODZ at ~12°S during October 2015 would have prevented the build-up of nitrite, as under these conditions denitrification shuts off and nitrification goes to completion.

Another notable difference between December 2012 and October 2015 at ~12°S is the lower N:P ratios in the upper 150 m during 2012 vs. 2015 (Fig. S7e). Again, higher oxygen concentrations in the upper 150 m during 2015 El Nino probably reduced the impact of fixed N loss processes on the N:P signatures of near-surface waters. These results imply that El Nino conditions could, at least, partially alleviate phytoplankton N limitation due to the reduction in the magnitude of denitrification. While it is beyond

the scope of the focus of this study, it would be interesting to examine whether this increase in N:P ratios during the 2015 El Nino impacted the phytoplankton communities within this region. The expectation that they may have impacted the phytoplankton communities is certainly reasonable (Rousseaux and Gregg, 2012), as Hauss et al. (2012) observed an increase in diatom biomass when the $NO_3^-:PO_4^{3-}$ ratios of water collected from the Peruvian upwelling region were increased.

The results from ~12°S shelf water indicate that upwelling of oxygen-poor water was still continuing in October 2015 at 12°S in the near surface layer, despite the enhanced SST anomaly related to El Niño. Below the oxycline, however, El Niño conditions were developing. The observations west of 77°48'W in the upper 75 m that oxygen as well as temperature (not shown) were lower in October 2015, may be





related to a stronger poleward flow which has been shown to be a characteristic of El Niño events (Strub et al., 1998).

At ~14°S at the easternmost station near the shelf, the oxygen distribution is quite similar for December 2012 and October 2015 (Supplement Fig. S8), indicating non-El Niño oxygen-poor upwelling near the

5 shelf. West of 77°W, the isopycnals are deeper in October 2015 related to a deeper thermocline with warmer water in the upper 100 m (not shown) and higher oxygen in October 2015 compared to December 2012. Similar to ~12°S the nutrient distribution shows higher nitrate and lower phosphate and silicate in October 2015 compared to December 2012 at ~14°S. This again shows a developing El Niño situation.

At ~16°S the oxygen concentrations at the shelf were lower in October 2015 than in December 2012 (Supplement Fig. S9), indicating similar as at ~14°S non-El Niño upwelling close to the shelf. The higher oxygen near the shelf in December 2012 was probably related to an unusual distribution related to an eddy located near the ~16°S section (e.g. Stramma et al., 2013; Czeschel et al., 2015). The SST at ~14°S and ~16°S was lower in October 2015 than in December 2012, hence the slight increase of SST

by El Niño did not compensate the typical seasonal SST signal. Different to the sections at ~9°S, ~12°S and ~14°S, at ~16°S the density distribution below the thermocline did not shift to higher densities in October 2015 which shows that the El Niño influence at 16°S was the weakest of the 4 shelf sections.

In summary, the temperature, salinity and oxygen measurements all indicate that during October 2015 the El Niño was strongest along our northern transects and weakest along our southern transects. This

was also apparent in the nutrient properties between the northern and southern portions of our study region. As outlined above, at 12°S the N:P ratio was higher and nitrite concentrations were lower during October 2015 when compared to the non-El Niño period of December 2012, both of which point to a reduction in the magnitude of denitrification. When comparing the differences between coastal nutricline N:P ratios and nitrite concentrations along the coast, we found that the differences between

October 2015 and December 2012 decreased between 12°S and 14°S, and again between 14°S and 16°S (data not shown). This again highlights the potential for El Niño events to impact N loss processes and upper water column biogeochemistry.

## 6 Conclusion

In this study, hydrographic measurements from a cruise to the eastern tropical Pacific in October 2015 were used to investigate the signal of the strong 2015 El Niño in the water mass distribution and in the EUC in comparison to measurements from the years 1993, 2009 and 2012. An increase of temperature from the surface to 350 m depth, and salinity in the 40 to 350 m depth layer appeared at the equator east of the Galapagos Islands at 85°30'W in October 2015. The warmer temperature led to lower densities despite the concurrent influence of the salinity increase on density. In October 2015, nitrate, phosphate and silicate concentrations were all lower in the upper 200 m when compared with previous non-El Nino periods, however, higher oxygen concentrations, which are characteristic of El Niño events, were only located between 40 and 130 m at the equator. Except for an oxygen increase in the upper ~60 m at 2°30'S, no obvious large vertical oxygen increase appeared at 1°N and 2°30'S at 85°30'W. This weak oxygen increase at and near the equator might be related to the weak EUC, which would else be expected to bring oxygen richer water eastwards.

Due to the influence of seasonal and El Niño signals, the velocity and transport observations of the EUC east of the Galapagos Islands were quite variable in the direct velocity measurements in different years. As previously observed in the central and western Pacific, and as predicted from model simulations, the EUC at the equator almost disappeared with a transport of only 0.02 Sv between 1°S and 1°N in October 2015 related to the El Niño conditions. Although weak, the EUC had shifted southward with a transport of 0.78 Sv between 2°30'S and 1°S in October 2015. These observations are in agreement with the predicted weakening and southward shift of the EUC in model results for El Niño periods (Montes et al., 2011). According to earlier observations, the disappearance of the EUC in the eastern Pacific seems to be related mainly to strong El Niño events. For the very strong 1982/83 El Niño, a disappearance of the EUC was described for the eastern Pacific (Halpern, 1997), whereas for the strong 1997/98 El Niño the EUC disappeared over all longitudes (Izumo, 2005). In contrast, during the moderate El Niños of 1986/87 and 1991/92 a disappearance was described in the western and central Pacific, but only a weakening in the eastern Pacific (McPhaden et al., 1990; McPhaden and Hayes, 1990; Izumo, 2005; Kessler and McPhaden, 1995; Seidel and Giese, 1999).

Four hydrographic sections near the Peruvian shelf between ~9°S and ~16°S had different El Niño related signals in October 2015. At ~9°S there was a large SST increase, and we observed upwelling of lighter water which was both warmer and more oxygenated, all of which are characteristic upwelling features of El Niño events. Between 12°S and 16°S the SST increase in October 2015 was weaker than at 9°S, and at the easternmost stations near the Peruvian shelf at ~12°S, ~14°S and ~16°S cold and oxygen-poor water was upwelled as during regular upwelling conditions. However, west of the easternmost stations El Niño type changes were observed below the thermocline and oxycline which weakened southward.

The 2015 El Niño started strong early in the year, and by October 2015 had an ONI index similar to earlier major El Niño events. The water characteristics at 85°30'W at the equator and EUC variability and upwelling at ~9°S also indicated that a strong EP El Niño had developed. However, at 1°N and 2°30'S at 85°30'W and at the sections near the shelf between 12°S and 16°S, the El Niño influence was still weak. To this end, the EUC clearly indicated a strong EP El Niño, while the distribution of hydrographic parameters, oxygen and nutrients indicated a transition period from regular to El Niño conditions progressing southward along the Peruvian shelf. Despite the strong 2015 El Niño, the shift to El Niño distribution in the eastern Pacific was surprisingly slow. As the ONI index increased to the end of 2015, we expect that the El Niño conditions were strengthening in the eastern Pacific after the cruise in October 2015. Measurements carried out by IFREMER with a Glider at about 8°S off Peru between 7 November and 17 December 2015 showed an increase in temperature and oxygen and a decrease in density at ~100 m when compared to October 2015, thus confirming the expected strengthening of the El Niño conditions.

(http://www.ego-network.org/dokuwiki/doku.php?glider=tenuse%2CEL_Nino&posti=0&postj=position_zoom0&pposti=0&ppostj=scaptemperature&hchk=&defsct=default_scatter)

**A Supplement is related to this article.**





*Author contributions.* L. Stramma and T. Fischer conceived the study, wrote the manuscript and carried out the ADCP and hydrographic measurements on the RV Sonne cruise in 2015 as well as on some of

the RV Meteor cruises. D.S. Grundle was co-chief scientist of the RV Sonne cruise in October 2015, organized the nutrient sampling and interpreted the nutrient data. G. Krahmann calibrated the RV Meteor and RV Sonne CTD data and interpreted the hydrographic data. H.W. Bange was chief scientist on RV Meteor in December 2012, he was responsible for the nutrient measurements on this cruise and interpreted the nutrient data. C.A. Marandino was chief scientist on the RV Sonne cruise in October

2015. All authors discussed and modified the manuscript.

*Acknowledgements.* The Deutsche Forschungsgemeinschaft (DFG) provided support as part of the "Sonderforschungsbereich 754: Climate-Biogeochemistry Interactions in the Tropical Ocean" and for the RV Meteor cruises. The Bundesministerium für Bildung und Forschung (BMBF) supported this

study as part of the project SOPRAN (03F0611A, 03F0662A) and through funding of the RV Sonne cruise in October 2015 (03G0243A). We thank the captains and crews of the RV Meteor and RV Sonne cruises for their help, R. Czeschel for helpful comments on the graphic software, T. Steinhoff for co-organizing the RV Sonne cruise, and M. Lohmann and H. Campen for the oxygen and nutrient measurements.

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

**Table 1.** Time and geographical location of CTD data used in this study and the NINO 1+2 and ONI indices for the months of observation or for 2 months for measurements carried out at the end or beginning of a month listed in the tables (http://www.cpc.ncep.noaa.gov/data/indices/sstoi.indices) and (http://www.cpc.ncep.noaa.gov/products/analysis_monitoring/ensostuff/ensoyears.shtml).

| | Time | location | NINO 1+2 | ONI |
|---|---|---|---|---|
| 20 | 29.03.-31.03.1993 | 1°N-2°40'S | +0.65 March +0.97 April | +0.5 March +0.7 April |
| | 12.02.-13.02.2009 | 1°N-2°40'S | -0.11 | -0.7 |
| | 01.11.-03.11.2012 | 1°N-2°40'S | -0.11 Oct.  -0.38 Nov. | +0.4 Oct.  +0.2 Nov. |
| | 07.10.-08.10.2015 | 1°N-2°30'S | +2.52 | +2.0 |
| 25 | 06.12.-23.12.2012 | ~9°S-~16°S | -0.68 | -0.2 |
| | 10.10.-19.10.2015 | ~9°S −~16°S | +2.52 | +2.1 |





**Table 2.** Summed zonal positive (eastward) and negative (westward) ADCP transports in Sv ($10^6$ m$^3$ s$^{-1}$) in the equatorial channel at 85°50'W in March 1993, February 2009 and November 2012 and at 85°30'W in October 2015 as well as the related El Niño status. The velocity data were slightly smoothed and extrapolated to the surface.

| Time | 1°S – 1°N, 0 – 300 m | | 2°30'S – 1°S, 0 – 300 m | | El Niño status |
|---|---|---|---|---|---|
| 29.03.-31.03.1993 | 12.77 | -0.38 | 6.28 | -0.07 | early El Niño-like |
| 12.02.-13.02.2009 | 3.55 | -1.58 | 0.55 | -1.57 | late La Niña-like |
| 01.11.-03.11.2012 | 10.78 | -0.94 | 4.22 | -0.36 | neutral |
| 07.10.-08.10.2015 | 0.02 | -13.86 | 0.78 | -4.08 | El Niño |



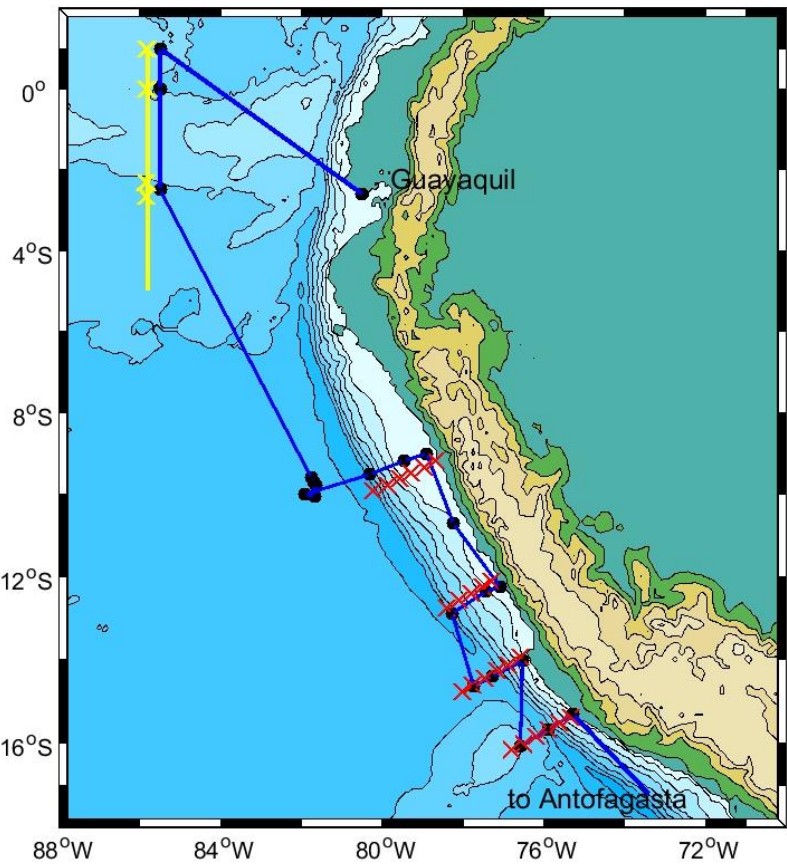

**Figure 1.** Cruise track (blue line) and CTD stations (black circles) of cruise RV *Sonne* from Guayaquil 5 October to Antofagasta 22 October 2015, as well as equatorial stations March 1993, February 2009 and November 2012 (yellow x), CTD sections off Peru December 2012 (red x) and ADCP sections 5 across the equator (yellow line). Topography (color) with depth/height contours in 1000 m intervals enhanced by the 200 m depth contour.





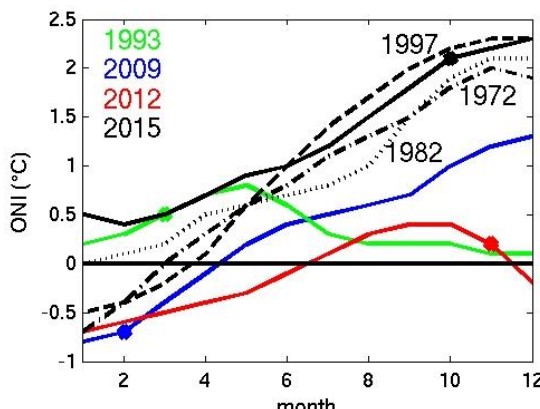

**Figure 2.** ONI index for the strong El Niño years 1972 (dash-dotted), 1982 (dotted), 1997 (dashed), and the years used here 1993 (green), 2009 (blue), 2012 (red) and 2015 (black line). The months of measurements used here are marked by coloured dots.

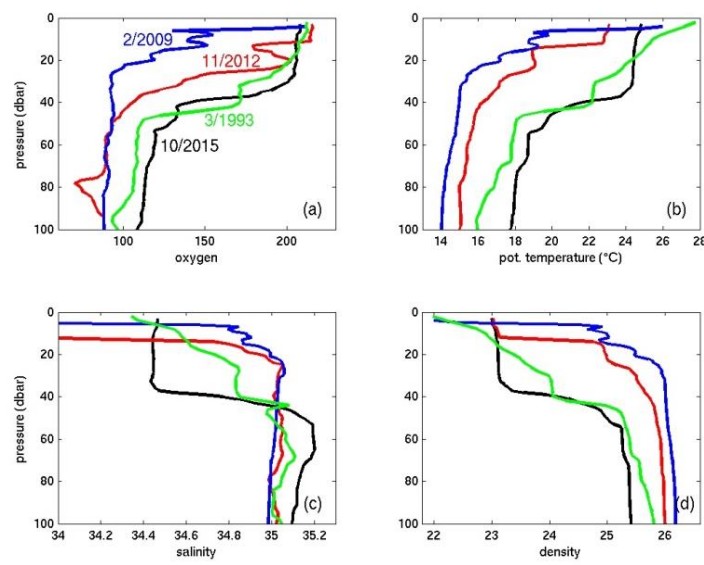

**Figure 3**. Upper 100 m profiles at the equator at 85°50'W for 30 March 1993 (green), 12 February 2009 (blue), 2 November 2012 (red) and at the equator at 85°30'W for 7 October 2015 (black) for **(a)** oxygen
10   in µmol kg$^{-1}$, **(b)** potential temperature in °C, **(c)** salinity and **(d)** potential density in kg m$^{-3}$.




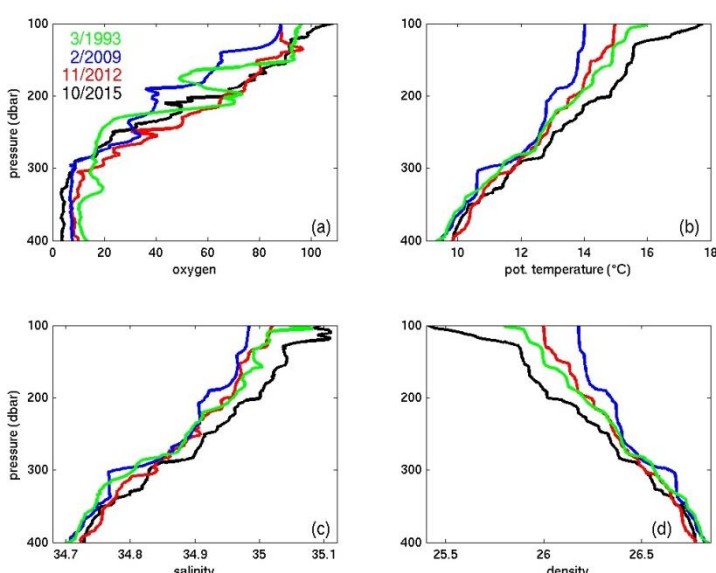

**Figure 4**. Same as Figure 3, but for 100 to 400 m depth.

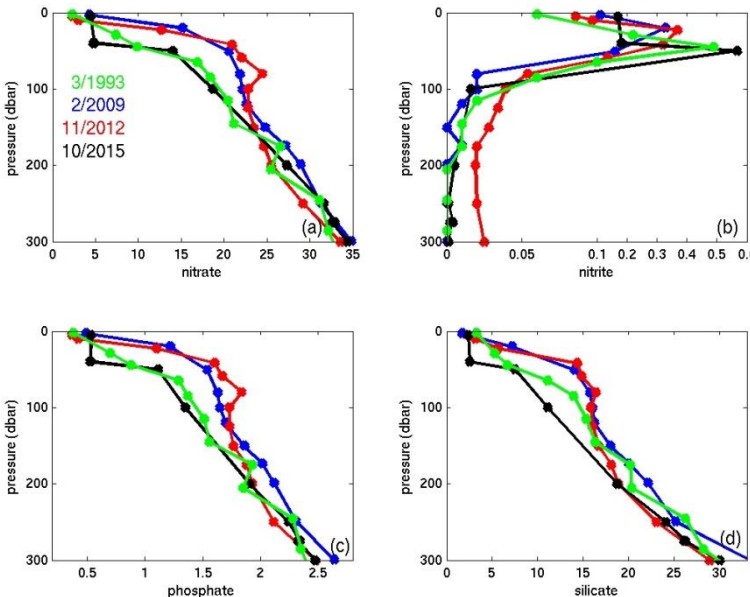

5 **Figure 5**. Upper 300 m profiles at the equator at 85°50'W for 30 March 1993 (green), 12 February 2009 (blue), 2 November 2012 (red) and at the equator At 85°30'W for 7 October 2015 (black) for **(a)** nitrate in μmol L$^{-1}$, **(b)** nitrite in μmol L$^{-1}$ (scale change at 0.1μmol L$^{-1}$), **(c)** phosphate in μmol L$^{-1}$ and **(d)** silicate in μmol L$^{-1}$.



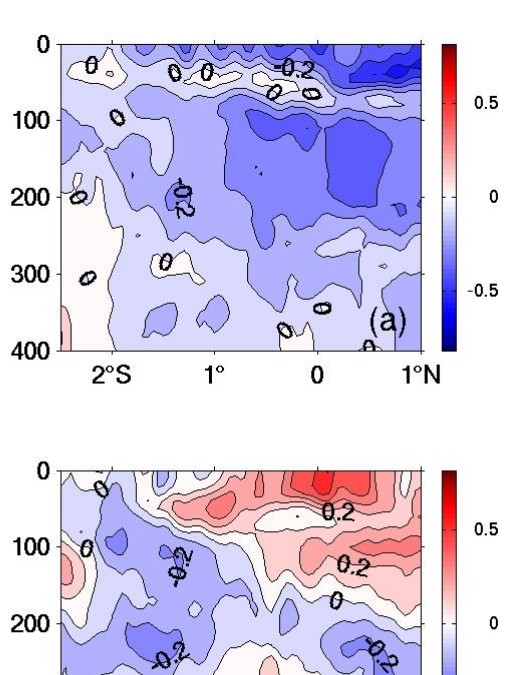

**Figure 6**. ADCP zonal **(a)** and meridional **(b)** velocity section (in m s$^{-1}$) on the diagonal section from
the Ecuadorian shelf at ~ 2°30'S to 1°N, 85°30'W (see Figure 1). Contour interval is 0.1 m s$^{-1}$.



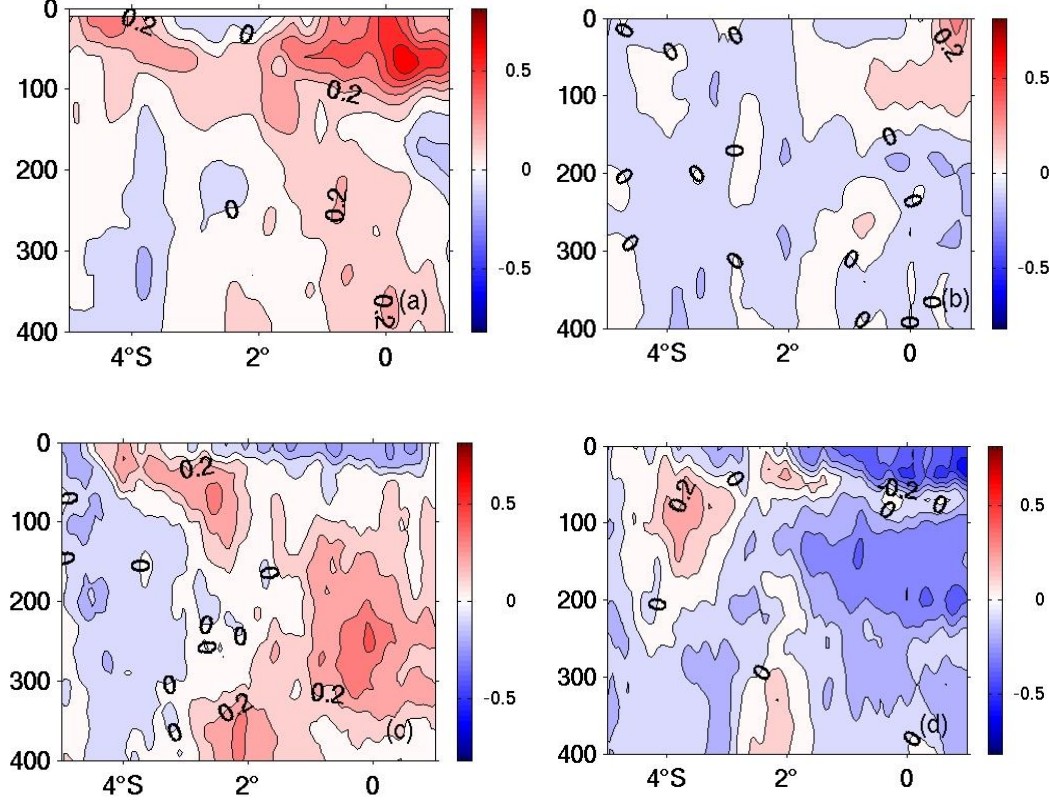

**Figure 7**. Zonal ADCP velocity sections (in m s$^{-1}$; positive eastward; contour interval 0.1 m s$^{-1}$) on the meridional path from 1°N to 5°S at 85°50'W **(a)** in March 1993, **(b)** in February 2009 **(c)** in November 2012 and **(d)** from 1°N to 2°30'S at 85°30'W and diagonal to 5°S 84°12'W in October 2015 (see Figure 1).



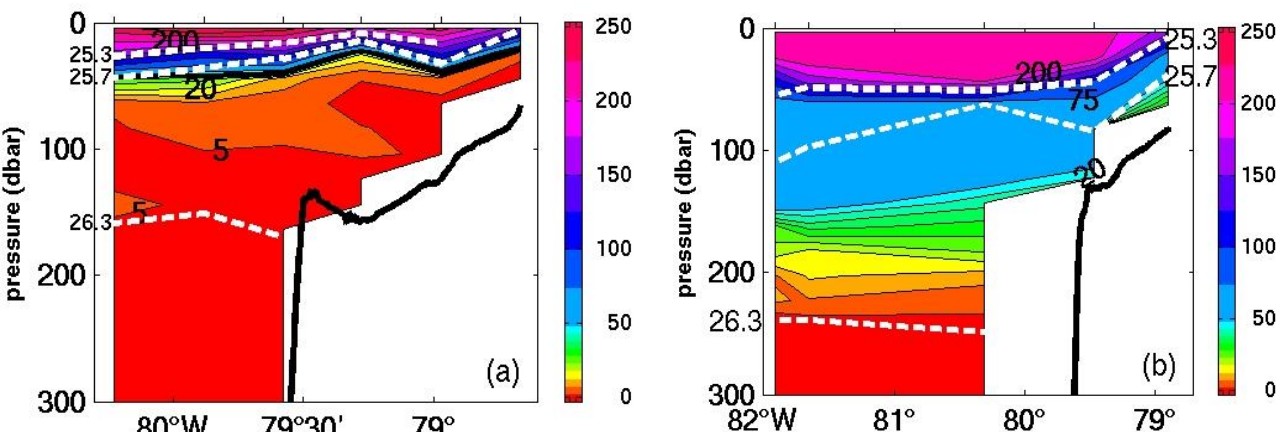

**Figure 8**. Oxygen section (color; in μmol kg$^{-1}$; same color-scale on both frames) at ~9°S off the Peruvian shelf for December 2012 **(a)** and October 2015 **(b)**. Three selected isopycnals are included as white dashed lines. Please note that the section in October 2015 reaches further west than in December 2012.

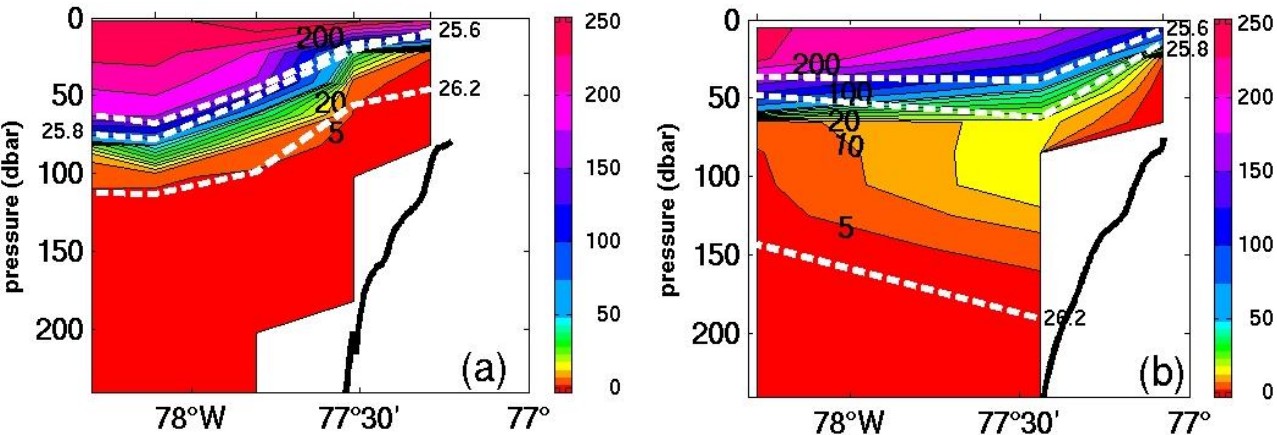

**Figure 9**. Oxygen section (color; in μmol kg$^{-1}$; same color-scale on both frames) at ~12°S off the Peruvian shelf for December 2012 **(a)** and October 2015 **(b)**. Three selected isopycnals are included as white dashed lines.