# Peer review of "Observed El Niño conditions in the eastern tropical Pacific in October 2015"

_Ocean Science, 2016_

## Referee Comment (RC1) · Anonymous Referee #1 · 27 Apr 2016

Review of the paper ÂńTransition to El Niño conditions in the eastern tropical Pacific in October 2015Âż by Stramma et al.

This paper documents the changes in water masses and circulation patterns in the equatorial and eastern south pacific from an oceanographic cruise which took place in october 2015 during El Niño (EN). Profiles and sections for temperature, salinity, oxygen, nutrients and ADCP current measurements are compared with those from previous cruises under different neutral, EN-like and La Niña-like conditions. The results show that the eastern flow associated to the EUC along the equator has greatly diminished in october 2015, and that temperature and oxygen have increased, while salinity has decreased. Near the Peru shore, cross-shore sections along the northern coast (9° and 12°S) display typical EN conditions with the upwelling of warmer, more oxygenated waters, while the EN patterns are not evidenced at lower latitudes (14° and

16°S).

General comment: Documenting EN conditions in this region of the Pacific is important particularly as there are not many measurements published in the literature during previous events. I found the paper well written and interesting to read. Most of my comments are minor.

Minor comments: Abstract: L23: transition to EN conditions west of the coastal sections is not well documented in the paper. Figures are not shown and it should be discussed how EN can be present offshore and not nearshore. What is the local process that could compensate the nearshore warming? Wind-driven upwelling? This should be investigated and discussed in more detail.

Introduction: P2L7: The denomination of Central EN or Modoki has been found in the literature previously to the publication of Dewitte et al., which is mainly a model study focused on the Peru region. I also think that other references (L10) should be cited to document the impact of EN off Chile (Ulloa et al., 2001 is an example but there must others more recent)

P2L19: Âńclimate modelling evidenceÂż: I would rather ponder this statement and write that climate models suggest that a doubling in the occurrence..

P3L20: Âńas a result of circulation changesÂż This is really vague. Could you be more specific?

P6L19: ÂńThe equatorial spreading of the thermoclineÂż This spreading of the thermocline is unclear to me. Is it zonal, meridional? Could you rephrase?

P6L25: Âń..intensifies from 16 to 15..Âż: is such a decrease of the mean oxygen concentration robust? This 1 micromole difference seems very small.

P7L10: The sentence is clumsy. I also do not understand the concluding sentence of the paragraph. Why should higher salinities be expected in the pycnocline during EN?

P8L6: What about local increase of precipitation and/or poleward displacement of the equatorial front (associated with the ITCZ) which separates fresh waters off Ecuador from more saline waters off Peru?

P8L21: higher PNMduring EN. I understand the deepening of the peak due to the deeper pycnocline but not the more intense PNM. How can it be explained? could this be due to an increase in denitrification?

P9L6: A citation of EN years considered in Czeschel et al. 2012 would be useful here.

P9L16: I would rather say an Âńocean modelÂż than a climate model which refers more often to ocean-atmosphere coupled models.

P10L2: Âńin Nov 1982Âż is repeated twice in the sentences

P10L11: in Fig 6, could you add Âńpositive eastward Âń for zonal velocities?

P11L18: ÂńNovember minimum at 95°WÂż. A reference is missing here and I do not see how this statement on the EUC at 95°W backs up the fact that it should be a reasonable estimate of the EUC in neutral EN conditions.

P12L5: the reference to Gutierrez et al. is misplaced. This sentence should be moved to L1 where SST are described. I am not sure Sydemann et al is really relevant here.

P12L16: I do not have a problem with citing Strub et al which is a review, but other papers should be cited as well (Halpern et al. 2001 , Enfield, 1981; Huyer et al., 1987)

P12L20: Here same remark as before, I think that Strub et al 1998 can be cited but other papers as well (which are cited by Strub in his review).

P12L26: ÂńReduction of the ODZ areaÂż: I do not understand how this area is computed. It seems more a vertical displacement of the OMZ than a reduction of its area. Could you clarify?

P13-16: I think there are too many figures in the supplementary here. Please reduce

them and add some of them to the paper section, as it is very difficult to follow without looking at the figures in the supplementary.

P14L3: could the lower nutrient concentrations in october 2015 be due to the seasonal variations of the nutricline?

P14L14: I would conclude this sentence by saying explicitly that this process may produce more nitrate. I think that is is important as one may think that all nutrient pools (phosphate, silicate AND nitrate) decrease during EN, which is not always the case from your observations.

P14L24: You suggest that diatom biomass may increase during EN due to the N:P increase. However both nitrate and phosphate concentration reduce strongly during EN, which should impact negatively phytoplankton growth more than the N:P increase. Previous studies have shown that the surface chlorophyll observed from satellite decreases during EN (Carr et al. 2002) and the ecosystem suffers dramatic changes during extreme EN (Barber and Chavez, 1983, Chavez et al. 2003).

P15L7: At 12°S, higher nitrate...Âż how can higher nitrate indicate a developing EN? This sounds contradictory.

P16L14: I think you should also mention the strong intraseasonal signals in the equatorial pacific in neutral periods, with the passage of upwelling and downwelling waves at intraseasonal time scales (Cravatte et al., 2003; Echevin et al. 2013). during a cruise, if sampling is performed during the passage of a downwelling wave in a nutral EN period, this might have some similarity with EN conditions.

P17L15: I do not understand the sentence. Please rephrase.

P17L18: Measurements are carried out by CNRS, IRD and IMARPE . IFRE-MER is the owner of the glider, which is part of the french national glider pool. Here is a link with more precise information (in french): https://www.ird.fr/toute-l-actualite/actualites/communiques-et-dossiers-de-presse/cp-2015/lancement-decienperu-projet-d-etude-des-impacts-d-el-nino-2015-2016-sur-l-ecosysteme-marin-du-perou

References: Ulloa et al., 2001. Evolution and biological efiects of the 1997-98 El Niño in the upwelling ecosystem ofi northern Chile, GEOPHYSICAL RESEARCH LETTERS, VOL. 28, NO. 8, PAGES 1591-1594, APRIL 15, 2001.

Halpern, 2002. Offshore Ekman transport and Ekman pumping off Peru during the 1997–1998 El Nino. GEOPHYSICAL RESEARCH LETTERS, VOL. 29, NO. 5, 10.1029/2001GL014097, 2002

Enfield, D. B. Thermally driven wind variability in the planetary boundary layer above Lima, Peru, J. Geophys. Res., 86, 2005 – 2016, 1981.

Huyer, A., R. L. Smith, and T. Paluszkiewicz. Coastal upwelling off Peru during normal and El Nino times, 1981 – 1984, J. Geophys. Res., 92, 14,297 – 14,307, 1987.

Carr, M.-E., P. T. Strub, A. C. Thomas, and J. L. Blanco, Evolution of 1996 – 1999 La Nina and El Nino conditions off the western coast of South America: A remote sensing perspective, J. Geophys. Res., 107(C12), 3236, doi:10.1029/2001JC001183, 2002.

Richard T. Barber and Francisco P. Chavez. Biological Consequences of El Niño, Science, New Series, Vol. 222, No. 4629 (Dec. 16, 1983), pp. 1203-1210, http://www.jstor.org/stable/1691793

Chavez et al. 2003. From Anchovies to Sardines and Back: Multidecadal Change in the Pacific Ocean, SCIENCE, VOL 299, 10 JANUARY 2003

Cravatte, S., Picaut, J., Eldin, G., 2003. Second and first baroclinic kelvin modes in the equatorial pacific intraseasonal timescales. J. Geophys. Res. 108 (C8), 3266, http: //dx.doi.org/10.1029/2002JC001511.

Echevin et al. 2013. Intraseasonal variability of nearshore productivity in the Northern Humboldt Current System: The role of coastal trapped waves, Continental Shelf

Research 73 (2014) 14–30

---

## Referee Comment (RC2) · Anonymous Referee #2 · 28 Apr 2016

The authors report on changes observed during the strong Eastern Pacific 'El Niño' event developed in early 2015, and compare this event to previous similar events, neutral, and one 'La Niña-like' event. As expected from models and previous observations, the SSTs increase in the eastern equatorial Pacific, the surface salinity drops, the EUC and other equatorial currents weaken, and the thermocline deepens. As a consequence, the concentration of nutrients decreases, the concentration of oxygen increases, and the OMZ shifts deeper. The effects of 'el Niño' do not spread all the way to the southern station by October 2015, off the coast in Peru.

I think that the paper is interesting and worth publishing, but I think that it would vastly improve if the writing is improved in some sections. There are several awkward sentences, and it is difficult to follow the flow of the paper if not an expert on the topic (especially the Introduction). I have minor comments mostly referred to organization

and flow.

The abstract and the Conclusions contain almost exactly the same information. I would suggest reducing the abstract substantially (it is too long), and keeping the Conclusions section as it is.

The Introduction is disorganized and difficult to follow. For example, changes due to climate change are discussed in Page2L5 and again in Page2L19. These should be discussed together. And before, the two types of el Nino should be explained. It makes it difficult to follow for non-experts. The description of indices in Page2L11 could go into Section 2 that could be called Datasets and Methods (or similar). If indices are discussed in the Intro, maybe not in the first paragraph.

I think that it is a good idea to move some of the Intro material into the mini-introductions in Section 4 and 5 (as it is), but the authors should avoid repetition with elements already described in the more general Introduction (Section 1). I suggest reducing the general Intro to avoid repetition, or put it all in the general Introduction (in which case you could include all the Seasonality changes together), or … (but avoiding the repetition of material).

Supplementary Fig. 1 should be a main Figure as it is crucial to understand the whole picture (maybe together with current Fig. 1?).

Could you somehow combine Fig. 3 and 4? (perhaps with irregular axis, with larger resolution from 0 to 100m).

Fig. 6 (add to the caption the date… October 2015, so that it's easier to follow without going back to text or Fig. 1)

Specific Comments:

Page1L11 "At the equator… October 2015" awkward phrasing

Page1L21 to L26 rephrase

Page2L4-5 rephrase Page2L5 "In contrast …." rephrase as for example 'There has been evidence.., different from the common cold tongue, Eastern Pacific El Nino events… ' This sentence should come after describing the two types of El Nino. Why is it relevant to talk about the Modoki type if none of the described events fall into this category? (or maybe I missed something). If it is significant, comment on it in the Conclusions?

Page3L5 avoid repetition of EUC description with text later on the draft.

Page3L18-25 "Oxygen increases as a result of circulation changes + explanation" This paragraph talking about oxygen needs more coherence and flow. The explanation about local winds is confusing and needs to be better linked with the following sentence (deeper thermocline) as it is counterintuitive. More upwelling but deeper thermocline? Etc…

Are all the changes in oxygen due to circulation or are any of these due to reduced primary production (L26)?

Page5L26 rephrase the sentence starting with ' Different… ' (suggestion: The SST distribution in fall 2015 shows a strong and prominent SST increase along Central America and in … that differs from the typical EP el Nino distribution. )

Page6L18 "The equatorial spreading of the thermocline… " rephrase and also explain what spreading of thermocline means.

Page7L8-11 rephrase paragraph? Not clear why salinity higher during EN events.

Page8L4 "In October 2015… salinity lower because reduced equatorial upwelling…" and because increased precipitation?

Page8L8 "In the profiles…" rephrase

Page9L3-8 What about March 1993? Oxygen is also high only in the high 60m… I would suggest to rephrase it so that it is easier to follow the author's logics, and more

clear. For example: The oxygen concentration was slightly higher only in the upper 60 m for both El Nino events in March 1993 and in October 2015 compared to ... . However, earlier selected measurements... showed a clear oxygen increase to a depth of 350m, hence we conclude that el Nino influence on the water mass distribution. ... .

Page9L20 "The modeled EUC transports..." are you giving some examples from OGCMs in the sentence before, or these are all the OGCMs used? Make it clear...

Page9L23 somewhat repeated information from the General Intro?

Page10L6-9 this segment belongs to intro.

From Page13 onwards the writing flows much better...

Page14L5-14 I don't understand why nitrate was lower (contrary to El Nino expectations). Clarify in the text?

End of page 15 You could move the summary of this sub-section (L18 onwards) into the conclusions.

---

## Author Comment (AC1) · 6 Jun 2016

Transition toObserved El Niño conditions in the eastern tropical Pacific in October 2015

Lothar Stramma1, Tim Fischer1, Damian S. Grundle1, Gerd Krahmann1, Hermann W. Bange1 and Christa A. Marandino1 1GEOMAR Helmholtz Centre for Ocean Research Kiel, Düsternbrooker Weg 20, 24105 Kiel, Germany

Michael McPhaden (NOAA) had a look at the original OSD manuscript and mentioned that the title is confusing. Therefore we modified the title as marked above. He also proposed to give a brief opening paragraph about why it is important to study El Nino and as both reviewers especially reviewer 2 proposed to modify the introduction we introduced now an opening paragraph.
[Figure]

Reviewer #1: Review of the paper Â'nTransition to El Niño conditions in the eastern tropical Pacific in October 2015ÂËŹz by Stramma et al. This paper documents the changes in water masses and circulation patterns in the equatorial and eastern south pacific from an oceanographic cruise which took place in october 2015 during El Niño (EN). Profiles and sections for temperature, salinity, oxygen, nutrients and ADCP current measurements are compared with those from previous cruises under different neutral, EN-like and La Niña-like conditions.The results show that the eastern flow associated to the EUC along the equator has greatly diminished in october 2015, and that temperature and oxygen have increased, while salinity has decreased. Near the Peru shore, cross-shore sections along the northern coast (9_ and 12_S) display typical EN conditions with the upwelling of warmer, more oxygenated waters, while the EN patterns are not evidenced at lower latitudes (14_ and 16_S). General comment: Documenting EN conditions in this region of the Pacific is important particularly as there are not many measurements published in the literature during previous events. I found the paper well written and interesting to read. Most of my comments are minor.

Answer to reviewer 1: We thank both reviewers for the helpful comments, which helped to improve the manuscript during the revision. We modified the manuscript as explained below in the detailed comments.

Reviewer #1: Minor comments: Abstract: L23: transition to EN conditions west of the coastal sections is not well documented in the paper. Figures are not shown and it should be discussed how EN can be present offshore and not nearshore. What is the local process that could compensate the nearshore warming? Wind-driven upwelling? This should be investigated and discussed in more detail.

Answer to reviewer 1: From our data we can't decode the processes responsible for the transition to EN west of the coastal stations. A possible explanation is the upwelling of left-over water from the pre El Nino period at the coastal station and the influence of the Peru Coastal Undercurrent which intensifies and shoals during El Nino at the stations west of the shelf station. Another explanation is that the observed transitional

feature of normal conditions near-shore and El Nino conditions offshore is probably a consequence of the cross-shore pattern in vertical velocity during upwelling. The near-shore vertical velocity is to be expected substantially larger than the offshore vertical velocity (Fennel, 1999). A downwelling Kelvin wave could then neutralize the weak offshore upwelling and bring down the thermocline, while near-shore the strong upwelling would hardly weaken and for some time still bring up remnants of cold oxygen-poor water, until supplies feed from the offshore warmer and oxygen replete waters. The wind field would not need to change in order to produce this transition pattern in hydrography. These explanations are now described in the revised manuscript and the former Figure S7 in now included in the main text as Figure 11.

Reviewer #1: Introduction: P2L7: The denomination of Central EN or Modoki has been found in the literature previously to the publication of Dewitte et al., which is mainly a model study focused on the Peru region. I also think that other references (L10) should be cited to document the impact of EN off Chile (Ulloa et al., 2001 is an example but there must others more recent)

Answer to reviewer 1: Reviewer 2 questioned to discuss the Modoki EN, as it is not investigated in our data set, nevertheless we like to mention its existence, but did not expand it to a longer discussion, we just added a reference to a Nature overview paper (Ashok and Yamagata, 2009). We added the proposed Ulloa et al. paper 2001 as an example for the influence of El Niño also on the Chilean region, however as the introduction should be shortened and we don't investigate the region off Chile and we did not expand this discussion further.

Reviewer #1: P2L19: Â′nclimate modelling evidenceÂËŹz: I would rather ponder this statement and write that climate models suggest that a doubling in the occurrence..

Answer to reviewer 1: Text was rewritten as proposed using 'suggest' instead of 'evidence'.

Reviewer #1: P3L20: Â′nas a result of circulation changesÂËŹz This is really vague.

[Figure]

Could you be more specific?

Answer to reviewer 1: This text on circulation changes was a first information on El Nino related changes for the reference to Helly and Levin 2004. As the processes involved are described in more detail in the manuscript, 'as a result of circulation changes' was removed here.

Reviewer #1: P6L19: Â'nThe equatorial spreading of the thermoclineÂËŹz This spreading of the thermocline is unclear to me. Is it zonal, meridional? Could you rephrase?

Answer to reviewer 1: This is meridional spreading and now 'meridional' is included and the reference to Johnson et al. 2002 was included.

Reviewer #1: P6L25: Â' n..intensifies from 16 to 15..ÂËŹz: is such a decrease of the mean oxygen concentration robust? This 1 micromole difference seems very small.

Answer to reviewer 1: Right, this change in oxygen of the ODZ core listed by Paulmier and Ruiz.Pino is removed from the text.

Reviewer #1: P7L10: The sentence is clumsy. I also do not understand the concluding sentence of the paragraph. Why should higher salinities be expected in the pycnocline during EN?

Answer to reviewer 1: This part was rewritten, removing the information on the late El Nino phase, which is not investigated in our manuscript. Instead the reason for the higher salinity in the pycnocline due to northward progression of the salinity maximum of the South Pacific Tropical Water under the modified current bands during El Nino is explained.

Reviewer #1: P8L6: What about local increase of precipitation and/or poleward displacement of the equatorial front (associated with the ITCZ) which separates fresh waters off Ecuador from more saline waters off Peru?

Answer to reviewer 1: Right, the higher precipitation will be the major contribution and is now mentioned before the equatorial upwelling. Some literature shows a very minor shift of the ITCZ in this region (e.g. Chen and Lin 2005, Monthly Weather Review), hence we did not mention it here.

Reviewer #1: P8L21: higher PNM during EN. I understand the deepening of the peak due to the deeper pycnocline but not the more intense PNM. How can it be explained? could this be due to an increase in denitrification?

Answer to reviewer 1: The higher nitrite concentrations in the PNM during EN are most probably resulting from (i) high nitrite conc. in the PNM of the SPTW which have been transported towards the equator and (ii) higher ammonium oxidation rates due to a reduced photo inhibition at deeper water depths during EN. Denitrification only occurs at suboxic/anoxic conditions. Therefore, the occurrence of denitrification is very unlikely because of the relatively high ambient O2 concentrations. We modified the text accordingly.

Reviewer #1: P9L6: A citation of EN years considered in Czeschel et al. 2012 would be useful here.

Answer to reviewer 1: The measurements are from the El Nino years 1982/83 and this information is now mentioned in the text.

Reviewer #1: P9L16: I would rather say an Â'nocean modelÂ̈Źz than a climate model which refers more often to ocean-atmosphere coupled models.

Answer to reviewer 1: In the title of the Cravatte et al. paper it is called 'climate model' and in the abstract it is named 'climate-type OGCM', but nevertheless we changed it to 'ocean model' to avoid misunderstanding.

Reviewer #1: P10L2: Â' nin Nov 1982Â̈Źz is repeated twice in the sentences

Answer to reviewer 1: The first reference to November 1982 in this sentence was removed.

Reviewer #1: P10L11: in Fig 6, could you add Â′ npositive eastward Â′n for zonal velocities?

Answer to reviewer 1: We added the velocity flow direction of both components as well as the date of measurements to the figure caption.

Reviewer #1: P11L18: Â′nNovember minimum at 95_WÂËŹz. A reference is missing here and I do not see how this statement on the EUC at 95_W backs up the fact that it should be a reasonable estimate of the EUC in neutral EN conditions.

Answer to reviewer 1: This transport refers to a reference by Johnson et al. 2002, which was listed earlier in the manuscript. This reference is now included also here.

Reviewer #1: P12L5: the reference to Gutierrez et al. is misplaced. This sentence should be moved to L1 where SST are described. I am not sure Sydemann et al is really relevant here.

Answer to reviewer 1: The sentence was moved to the beginning of the paragraph where the SST is described as proposed. The reference to Sydemann was removed.

Reviewer #1: P12L16: I do not have a problem with citing Strub et al which is a review, but other papers should be cited as well (Halpern et al. 2001 , Enfield, 1981; Huyer et al., 1987)

Answer to reviewer 1: For this text part Halpern 2002, Enfield 1981 and Huyer et al., 1987 were included.

Reviewer #1: P12L20: Here same remark as before, I think that Strub et al 1998 can be cited but other papers as well (which are cited by Strub in his review).

Answer to reviewer 1: Again, for this text part Halpern 2002, Enfield 1981 and Huyer et al., 1987 were included.

Reviewer #1: P12L26: Â′nReduction of the ODZ areaÂËŹz: I do not understand how this area is computed. It seems more a vertical displacement of the OMZ than a reduction of its area. Could you clarify?

Answer to reviewer 1: The ODZ was defined for dissolved oxygen < 0.5 ml/l and the area considered 6°-20°S. According to the paper by Helly and Levin (2004) there could be a vertical displacement with a depression by 100 m and the area could be reduced from 77,000 to 30,000 km2. This information is now added.

Reviewer #1: P13-16: I think there are too many figures in the supplementary here. Please reduce them and add some of them to the paper section, as it is very difficult to follow without looking at the figures in the supplementary.

Answer to reviewer 1: We moved the supplementary figures S1 and S7 to the main text as they are important to understand the observations. The other supplementary figures are of interest for readers with specific detailed interest and we left them in the supplement.

Reviewer #1: P14L3: could the lower nutrient concentrations in october 2015 be due to the seasonal variations of the nutricline?

Answer to reviewer 1: We do not see any evidence to attribute this decrease in nutrient concentrations to normal seasonal variations. For example, the recent discussion paper by Graco et al. (http://www.biogeosciences-discuss.net/bg-2015-567/bg-2015-567.pdf) shows a nutrient concentration time series from 1996 to 2010. The differences between October and December are small for most of the years presented and for all nutrients measurements. Unfortunately, the time series presented does not include either year presented in our manuscript, but we conclude that there is no distinct pattern between October and December since 1996.

Reviewer #1: P14L14: I would conclude this sentence by saying explicitly that this process may produce more nitrate. I think that is is important as one may think that all nutrient pools (phosphate, silicate AND nitrate) decrease during EN, which is not always the case from your observations.

Answer to reviewer 1: We have now wrapped up the sentence by clearly pointing out that complete nitrification will result in more nitrate.

Reviewer #1: P14L24: You suggest that diatom biomass may increase during EN due to the N:P increase. However both nitrate and phosphate concentration reduce strongly during EN, which should impact negatively phytoplankton growth more than the N:P increase. Previous studies have shown that the surface chlorophyll observed from satellite decreases during EN (Carr et al. 2002) and the ecosystem suffers dramatic changes during extreme EN (Barber and Chavez, 1983, Chavez et al. 2003).

Answer to reviewer 1: In this section we were comparing data from 2012 with data from 2015 (now Fig. 11). While the reviewer is correct that a reduction in both nitrate and phosphate could negatively impact phytoplankton growth even if the N:P ratio was higher, we did not see a decrease in nitrate during our October 2015 cruise. Instead, we found that both nitrate and N:P was higher, which, as we suggest, could increase phytoplankton biomass.

Reviewer #1: P15L7: At 12_S, higher nitrate...ÂËŹz how can higher nitrate indicate a developing EN? This sounds contradictory.

Answer to reviewer 1: As previously outlined, higher oxygen concentrations during El Nino would result in less denitrification and more complete nitrification, which in turn would result in more nitrate. We have now added text to the end of this section to make this more clear.

Reviewer #1: P16L14: I think you should also mention the strong intraseasonal signals in the equatorial pacific in neutral periods, with the passage of upwelling and down-welling waves at intraseasonal time scales (Cravatte et al., 2003; Echevin et al. 2013). during a cruise, if sampling is performed during the passage of a downwelling wave in a nutral EN period, this might have some similarity with EN conditions.

Answer to reviewer 1: The influence of intraseasonal signals is now mentioned and the

references Cravatte et al. 2003 and Echevin et al. 2014 are included.

Reviewer #1: P17L15: I do not understand the sentence. Please rephrase.

Answer to reviewer 1: This sentence was rewritten separating the equatorial observations and the observations off the South American continent.

Reviewer #1: P17L18: Measurements are carried out by CNRS, IRD and IMARPE . IFREMER is the owner of the glider, which is part of the french national glider pool. Here is a link with more precise information (in french): https://www.ird.fr/toutel- actualite/actualites/communiques-et-dossiers-de-presse/cp-2015/lancement-de- cienperu-projet cienperu-projet-d-etude-des-impacts-d-el-nino-2015-2016-sur-l-ecosysteme-marindu- perou

Answer to reviewer 1: The text was modified according to your information on the groups carrying out the measurements and the web-link was replaced.

---

## Author Comment (AC2) · 6 Jun 2016

Observed El Niño conditions in the eastern tropical Pacific in October 2015

Lothar Stramma1, Tim Fischer1, Damian S. Grundle1, Gerd Krahmann1, Hermann W. Bange1 and Christa A. Marandino1

1GEOMAR Helmholtz Centre for Ocean Research Kiel, Düsternbrooker Weg 20, 24105 Kiel, Germany

Michael McPhaden (NOAA) had a look at the original OSD manuscript and mentioned that the title is confusing. Therefore we modified the title as written above. He also proposed to give a brief opening paragraph about why it is important to study El Nino and as both reviewers especially reviewer 2 proposed to modify the introduction we

introduced now an opening paragraph.

Reviewer #2: The authors report on changes observed during the strong Eastern Pacific 'El Niño' event developed in early 2015, and compare this event to previous similar events, neutral, and one 'La Niña-like' event. As expected from models and previous observations, the SSTs increase in the eastern equatorial Pacific, the surface salinity drops, the EUC and other equatorial currents weaken, and the thermocline deepens. As a consequence, the concentration of nutrients decreases, the concentration of oxygen increases, and the OMZ shifts deeper. The effects of 'el Niño' do not spread all the way to the southern station by October 2015, off the coast in Peru. I think that the paper is interesting and worth publishing, but I think that it would vastly improve if the writing is improved in some sections. There are several awkward sentences, and it is difficult to follow the flow of the paper if not an expert on the topic (especially the Introduction). I have minor comments mostly referred to organization and flow.

Answer to reviewer 2: We thank both reviewers for the helpful comments, which helped to improve the manuscript during the revision. We modified the manuscript as explained below in the detailed comments.

Reviewer #2: The abstract and the Conclusions contain almost exactly the same information. I would suggest reducing the abstract substantially (it is too long), and keeping the Conclusions section as it is.

Answer to reviewer 2: The abstract was shortened and the conclusion extended by shifting the end of page 15 to the conclusions as you proposed.

Reviewer #2: The Introduction is disorganized and difficult to follow. For example, changes due to climate change are discussed in Page2L5 and again in Page2L19. These should be discussed together. And before, the two types of el Nino should be explained. It makes it difficult to follow for non-experts. The description of indices in Page2L11 could go into Section 2 that could be called Datasets and Methods (or similar). If indices are discussed in the Intro, maybe not in the first paragraph.

Answer to reviewer 2: The order of the presented background information in the introduction was rearranged and the paragraph on the indices moved to the methods and called now as proposed 'Data sets and methods'.

Reviewer #2: I think that it is a good idea to move some of the Intro material into the mini-introductions in Section 4 and 5 (as it is), but the authors should avoid repetition with elements already described in the more general Introduction (Section 1). I suggest reducing the general Intro to avoid repetition, or put it all in the general Introduction (in which case you could include all the Seasonality changes together), or : : : (but avoiding the repetition of material).

Answer to reviewer 2: We shortened the detailed information in the introduction and presented the detailed information in the later paragraphs.

Reviewer #2: Supplementary Fig. 1 should be a main Figure as it is crucial to understand the whole picture (maybe together with current Fig. 1?).

Answer to reviewer 2: We moved the Supplementary Fig. S1 to the main text as Figure 2 and renamed the other figures accordingly. The region of this new figure 2 for the eastern Pacific is too small to integrate Figure 1 in a Pacific-wide figure to see the details of the different data sets used.

Reviewer #2: Could you somehow combine Fig. 3 and 4? (perhaps with irregular axis, with larger resolution from 0 to 100m).

Answer to reviewer 2: As there are large differences of the scales of oxygen, temperature, salinity and density in the upper 100 m and 100 to 400 m, a combination of both figures would lead to strongly reduced visibility of the changes related to the different El Nino phases. Hence we prefer to keep the two figures separated.

Reviewer #2:

Fig. 6 (add to the caption the date: : : October 2015, so that it's easier to follow without going back to text or Fig. 1)

Answer to reviewer 2: We added the date of measurements as well as the velocity flow direction to the figure caption.

Reviewer #2: Specific Comments: Page1L11 "At the equator: : : October 2015" awkward phrasing

Answer to reviewer 2: This sentence was reorganized for better readability.

Reviewer #2: Page1L21 to L26 rephrase

Answer to reviewer 2: The shortening of the abstract as you proposed also led to a rephrasing of the former lines 21 to 26 on page 1.

Reviewer #2: Page2L4-5 rephrase Page2L5 "In contrast : : :." rephrase as for example 'There has been evidence.., different from the common cold tongue, Eastern Pacific El Nino events: : : ' This sentence should come after describing the two types of El Nino. Why is it relevant to talk about the Modoki type if none of the described events fall into this category? (or maybe I missed something). If it is significant, comment on it in the Conclusions?

Answer to reviewer 2: The sentence was rephrased and shifted behind the general description of the 'regular' El Niño as proposed. As the Modoki El Niño is now often mentioned we like to introduce it briefly for readers not familiar with this difference although we do not investigate it in our manuscript. As reviewer 1 proposed to describe the Modoki El Niño with more references, we added a reference to an overview paper in Nature, but we did not expand the discussion further.

Reviewer #2: Page3L5 avoid repetition of EUC description with text later on the draft.

Answer to reviewer 2: The information on the EUC was shortened in the introduction and the detailed information was shifted to the later paragraphs.

Reviewer #2: Page3L18-25 "Oxygen increases as a result of circulation changes + explanation" This paragraph talking about oxygen needs more coherence and flow.

The explanation about local winds is confusing and needs to be better linked with the following sentence (deeper thermocline) as it is counterintuitive. More upwelling but deeper thermocline? Etc: : :

Answer to reviewer 2: The information is added that the upwelling favorable winds could not produce the observed warming, but that warming is related to the downwelling Kelvin waves.

Reviewer #2: Are all the changes in oxygen due to circulation or are any of these due to reduced primary production (L26)?

Answer to reviewer 2: The additional influence of reduced primary production is now mentioned and a reference to Gutierrez et al. 2008 was added.

Reviewer #2: Page5L26 rephrase the sentence starting with ' Different: : : ' (suggestion: The SST distribution in fall 2015 shows a strong and prominent SST increase along Central America and in : : : that differs from the typical EP el Nino distribution. )

Answer to reviewer 2: The sentence was rephrased as proposed.

Reviewer #2: Page6L18 "The equatorial spreading of the thermocline: : : " rephrase and also explain what spreading of thermocline means.

Answer to reviewer 2: The sentence was from the original Johnson et al. 2002 paper. Now we included the information that the meridional spreading of the thermocline is related to the zonsl EUC velocity strength.

Reviewer #2: Page7L8-11 rephrase paragraph? Not clear why salinity higher during EN events.

Answer to reviewer 2: This part was rewritten, removing the information on the late El Nino phase, which is not investigated in our manuscript and instead the reason for the higher salinity in the pycnocline due to northward progression of the salinity maximum of the South Pacific Tropical Water under the modified current bands during El Nino is

explained.

Reviewer #2: Page8L4 "In October 2015: : : salinity lower because reduced equatorial upwelling: : :" and because increased precipitation?

Answer to reviewer 2: Right, the higher precipitation will be the major contribution and is now mentioned before the equatorial upwelling.

Reviewer #2: Page8L8 "In the profiles: : :" rephrase

Answer to reviewer 2: The part 'In the profiles to 400 m depth' was moved and the sentence rearranged.

Reviewer #2: Page9L3-8 What about March 1993? Oxygen is also high only in the high 60m: : : I would suggest to rephrase it so that it is easier to follow the author's logics, and more clear. For example: The oxygen concentration was slightly higher only in the upper 60 m for both El Nino events in March 1993 and in October 2015 compared to : : : . However, earlier selected measurements: : : showed a clear oxygen increase to a depth of 350m, hence we conclude that el Nino influence on the water mass distribution: : :. .

Answer to reviewer 2: We rewrote this text part as proposed.

Reviewer #2: Page9L20 "The modeled EUC transports: : :" are you giving some examples from OGCMs in the sentence before, or these are all the OGCMs used? Make it clear: : :

Answer to reviewer 2: Yes, these are OGCM examples from the ROMS model mentioned in the sentence before. This information is now included also in the sentence where the examples are given.

Reviewer #2: Page9L23 somewhat repeated information from the General Intro?

Answer to reviewer 2: The information on the EUC was shortened in the introduction and the sentence on p9 was removed as the major discussion is now in the conclusion

paragraph.

Reviewer #2: Page10L6-9 this segment belongs to intro.

Answer to reviewer 2: This text was shifted to the introduction.

Reviewer #2: From Page13 onwards the writing flows much better: : :

Answer to reviewer 2: Ok.

Reviewer #2: Page14L5-14 I don't understand why nitrate was lower (contrary to El Nino expectations). Clarify in the text?

Answer to reviewer 2: Nitrate was actually higher in October 2015 compared to December 2012, as opposed to being lower. Under El Nino conditions upwelling is reduced, and this prevents nutrients such as phosphate and silicate from building up in the mixed layer, however, nitrate and nitrite are different because their distributions are driven more by oxygen availability which regulates nitrification and denitrification. The higher oxygen concentrations during El Nino events reduces fixed N losses and results in higher nitrate concentrations. We have now added this explanation in the text.

Reviewer #2:

End of page 15 You could move the summary of this sub-section (L18 onwards) into the conclusions.

Answer to reviewer 2: This paragraph was moved to the conclusions as proposed.

---

## Author Comment (AC3) · 6 Jun 2016

The marked changes in the modified title of the manuscript are not visible in the reply text. The title should read now: 'Observed El Nino conditions in the eastern tropical Pacific in October 2015'

---

## Author Response (AR2)

**Dear Dr. Hoppema                14 June 2016**

**Thank you for accepting the manuscript for publication and taking your time for the evaluation of the manuscript text for final modifications. We note the changes we made in the copy of your comments below.**

**In addition to the requested changes we added a paragraph "Data availability" listing the references to the data used.**

**The author Damian Grundle is now employed at BIOS in Bermuda and we added for him the BIOS address for his institutional address.**

**With best wishes**
**Lothar Stramma**

**Topic Editor Decision: Publish subject to technical corrections**
(10 Jun 2016) by Dr. Mario Hoppema
Comments to the Author:
Dear Dr. Stramma and co-authors,

Thank you for the revision and resubmission of your manuscript. It is now accepted for publication in Ocean Science. Please have a look at some few technical issues which are listed below and correct these.

Answer: All comments are taken care of, see below.

P21 L24 "upwelling was modified" I don't quite understand what you mean here. Is it: the characteristics of upwelling were different?

Answer: Thank you, we modified the text as you proposed as 'characteristics of upwelling were different'.

P23 L 19 delete: more (rare is rare)

Answer: 'more' was deleted.

P24L1 … of Peru; however, for the EP El Niño … (substitute second: while) Twice "while" in one sentence is not nice.

Answer: 'while' was removed and the sentence split into two sentences.

P34 L14 Please use SI units for the oxygen concentration, or at least translate the boundary of 0.5 ml l-1 to xx µmol l-1 (or umol kg-1) and give both.

Answer: SI unit is included.

Table 1 and 2: Please comply with the date format of Ocean Science: 29-31 March 1993; 12-13 February 2009 (or 12-13 Feb); etc.

Answer: Tables 1 and 2 were modified to fit the OS Format.

Figure 4, 5, 6: I am not sure the axis labels will be readable. Please increase the letter size.

Answer: The axis labels were increased in size for the figures 4, 5, and 6 as well as for the supplement Figures S1, S2 and S3.

With best wishes
Mario Hoppema